# The origins and genetic interactions of *KRAS* mutations are allele- and tissue-specific

Joshua H. Cook [1,2,3,6], Giorgio E. M. Melloni [3,6], Doga C. Gulhan[3], Peter J. Park [3✉] & Kevin M. Haigis [1,2,4,5✉]

Mutational activation of *KRAS* promotes the initiation and progression of cancers, especially in the colorectum, pancreas, lung, and blood plasma, with varying prevalence of specific activating missense mutations. Although epidemiological studies connect specific alleles to clinical outcomes, the mechanisms underlying the distinct clinical characteristics of mutant *KRAS* alleles are unclear. Here, we analyze 13,492 samples from these four tumor types to examine allele- and tissue-specific genetic properties associated with oncogenic *KRAS* mutations. The prevalence of known mutagenic mechanisms partially explains the observed spectrum of *KRAS* activating mutations. However, there are substantial differences between the observed and predicted frequencies for many alleles, suggesting that biological selection underlies the tissue-specific frequencies of mutant alleles. Consistent with experimental studies that have identified distinct signaling properties associated with each mutant form of KRAS, our genetic analysis reveals that each *KRAS* allele is associated with a distinct tissue-specific comutation network. Moreover, we identify tissue-specific genetic dependencies associated with specific mutant *KRAS* alleles. Overall, this analysis demonstrates that the genetic interactions of oncogenic *KRAS* mutations are allele- and tissue-specific, underscoring the complexity that drives their clinical consequences.

[1] Department of Cancer Biology, Dana-Farber Cancer Institute, Boston, MA, USA. [2] Department of Medicine, Brigham & Women's Hospital, Harvard Medical School, Boston, MA, USA. [3] Department of Biomedical Informatics, Harvard Medical School, Boston, MA, USA. [4] Broad Institute, Cambridge, MA, USA. [5] Harvard Digestive Disease Center, Harvard Medical School, Boston, MA, USA. [6] These authors contributed equally: Joshua H. Cook, Giorgio E. M. Melloni. ✉email: peter_park@hms.harvard.edu; kevin_haigis@dfci.harvard.edu

Located at a critical signaling junction between extracellular growth receptors and pro-growth pathways, *KRAS* is one of the most commonly mutated genes in cancer[1,2]. However, it is frequently mutated in only a handful of cancers, with the highest frequencies in colorectal adenocarcinoma (COAD), lung adenocarcinoma (LUAD), multiple myeloma (MM), and pancreatic adenocarcinoma (PAAD). Importantly, the activating alleles found in *KRAS* vary substantially across cancers, indicating possible differences in signaling behavior of the mutant proteins that exploit the environment of the specific cellular context[3,4].

When mutated at one of its four hotspot codons—12, 13, 61, or 146—activated KRAS protein hyperactivates many downstream effector pathways, such as the MAPK and PI3K-AKT signaling pathways[1]. Previous studies have documented substantial differences in the biochemical and signaling properties of the common KRAS variants (reviewed by Miller et al.[5] and Li et al.[6]). KRAS normally operates as a molecular switch, activating downstream pathways when GTP-bound, but inactive when GDP-bound following the hydrolysis of the γ-phosphate. This reaction is catalyzed by GTPase-activating proteins (GAPs), while the exchange of the GDP for a new GTP is facilitated by guanine nucleotide exchange factors (GEFs)[7]. Activating *KRAS* mutations result in elevated engagement of downstream pathways by increasing the steady-state levels of GTP-bound KRAS. Specifically, mutations to codons 12, 13, and 61 reduce the rate of intrinsic and/or GAP-mediated hydrolysis, and mutations at 13 and 61, but not 12, also enhance the rate of nucleotide exchange[8,9]. Alternatively, 146 mutations do not alter the rate of GTP hydrolysis, but cause hyperactivation through an increased rate of GDP exchange[4,10–12]. Additional biochemical, structural, and signaling distinctions have been identified between different mutant alleles, including between those at the same amino acid position[4,8,13–20].

Likely as a consequence of their distinct properties, associations have been uncovered between the specific *KRAS* mutation status and therapeutic responses and clinical outcomes of cancer patients[3,6]. For instance, a retrospective meta-analysis suggested that COAD tumors with a *KRAS* G13D allele were sensitive to anti-EGFR therapies, a treatment generally discouraged for *KRAS*-mutant tumors[21]. It has recently been proposed, via computational and experimental means, that differential interaction kinetics between KRAS G13D and the Ras GAP NF-1 explain this effect[22–24]. Another example is that the *KRAS* G12D allele is associated with worse overall survival in advanced PAAD, when compared to patients with WT *KRAS*, *KRAS* G12R, or *KRAS* G12V[25]. Thus far, the hypothesis has been that the different biological properties of the mutant *KRAS* alleles are the cause of these clinical distinctions. However, it is also possible that allele-specific genetic interactions drive the varying clinical outcomes.

Understanding the heterogeneous properties of the *KRAS* alleles is essential to effectively treating *KRAS*-driven cancers. Here, we study the origins of *KRAS* mutations to assess the extent to which tissue-specific mutational processes determined the allelic distribution. We then construct comutation networks for each *KRAS* allele to identify different properties of the alleles. Finally, we analyze allele-specific genetic dependencies to explore potential therapeutic targets. Our analysis demonstrates that an allele-specific and tissue-specific analysis is necessary to fully understand the nature of the most potent oncogenes.

## Results

### *KRAS* alleles are non-uniformly distributed across cancers.
This study utilized publicly available sequencing data from COAD, LUAD, MM, and PAAD. There were whole-exome or genome data available for 1536 COAD (including 256

hypermutated samples), 891 LUAD, 1,201 MM, and 1395 PAAD samples. In addition, there were targeted-sequencing data available for 3329 COAD (including 464 hypermutated samples), 4160 LUAD, 61 MM, and 919 PAAD samples. More information on the data is available in "Methods" and Supplementary Data 1 and 2.

Across all the alleles, *KRAS* was most frequently mutated in PAAD (86%), followed by COAD (41%), LUAD (35%), and MM (22%; Fig. 1a). At the allele level, most mutations by single-nucleotide substitutions occurred at one of four "hotspot" codons: 12, 13, 61, and 146 (Fig. 1b and Supplementary Data 3). Glycine 12 and 13 can be transformed to six different amino acids (A, C, D, R, S, and V) through single-nucleotide changes in the first two guanine residues. Glutamine 61 can be mutated to six other amino acids (E, H, K, L, P, and R) and a stop codon via a single-nucleotide mutation. Alanine 146 can become one of six other amino acids (E, G, P, S, T, and V) from mutations to a single nucleotide.

Of these hotspots, codon 12 mutations accounted for 72.2% of all mutations in the dataset, followed by codon 13 (9.8%), 61 (14.8%), and 146 (3.2%). Adjusting for the yearly incidence of each cancer, the distribution of mutations was 76.8, 11.4, 8.1, and 3.7% at codons 12, 13, 61, and 146, across the four cancers. Importantly, there was substantial variability of the alleles found at these hotspots across the four *KRAS*-driven cancers (Fig. 1b). For example, MM was the only cancer where a non-G12 allele, Q61H, was the most frequent. At codon 12, LUAD had an enrichment for G12C mutations. COAD had a unique enrichment of G13D and A146T alleles, while PAAD was distinct in its high frequency of G12R mutations.

### The *KRAS* alleles have different mutagenic origins.
One potential explanation for the distinct allelic frequencies across cancer types is that tissue-specific mutational processes determine the frequency distribution. To explore this hypothesis, we elucidated the active mutational processes in the tumor samples using mutational signatures[26] (Supplementary Data 4 and 5; the signature numbers refer to those in the catalog published by Alexandrov et al.[27]). Briefly, all single-nucleotide mutations can be represented by the combination of the six possible base substitutions (C > T, C > A, C > G, T > A, T > C, and T > G) and all possible 3′ and 5′ flanking bases. This composes a mutational spectrum with 96 different trinucleotide contexts. We computed the spectrum of mutational signatures in the whole-exome and whole-genome sequencing (WGS) data using nonnegative matrix factorization and measured in each sample using nonnegative least squares regression (see "Methods" and Supplementary Fig. 1a, b).

As expected, the distributions of the levels of each mutational signature were highly variable across tumor types. The most common in COAD, MM, and PAAD, were the "clock-like" single base substitution (SBS) signatsures SBS1 and SBS5 (Fig. 1c and Supplementary Fig. 1c), which are believed to accumulate with age[28]. LUAD was uniquely enriched for a mutational signature of exogenous cause, tobacco smoke carcinogens (SBS4). Within each cancer type, the relative abundance of the mutational signatures was generally consistent across tumor samples, regardless of the *KRAS* allele (Fig. 1c). One exception was for cancers with microsatellite instability (MSI), in which defective DNA mismatch repair and other related signatures dominated (Supplementary Fig. 1a, b). Some instances of differential mutational signature composition between tumor samples with different *KRAS* alleles were identified, though they tended to be differences in magnitude of the signatures, not their presence or absence (Supplementary Fig. 2). Thus, for each cancer, the allelic

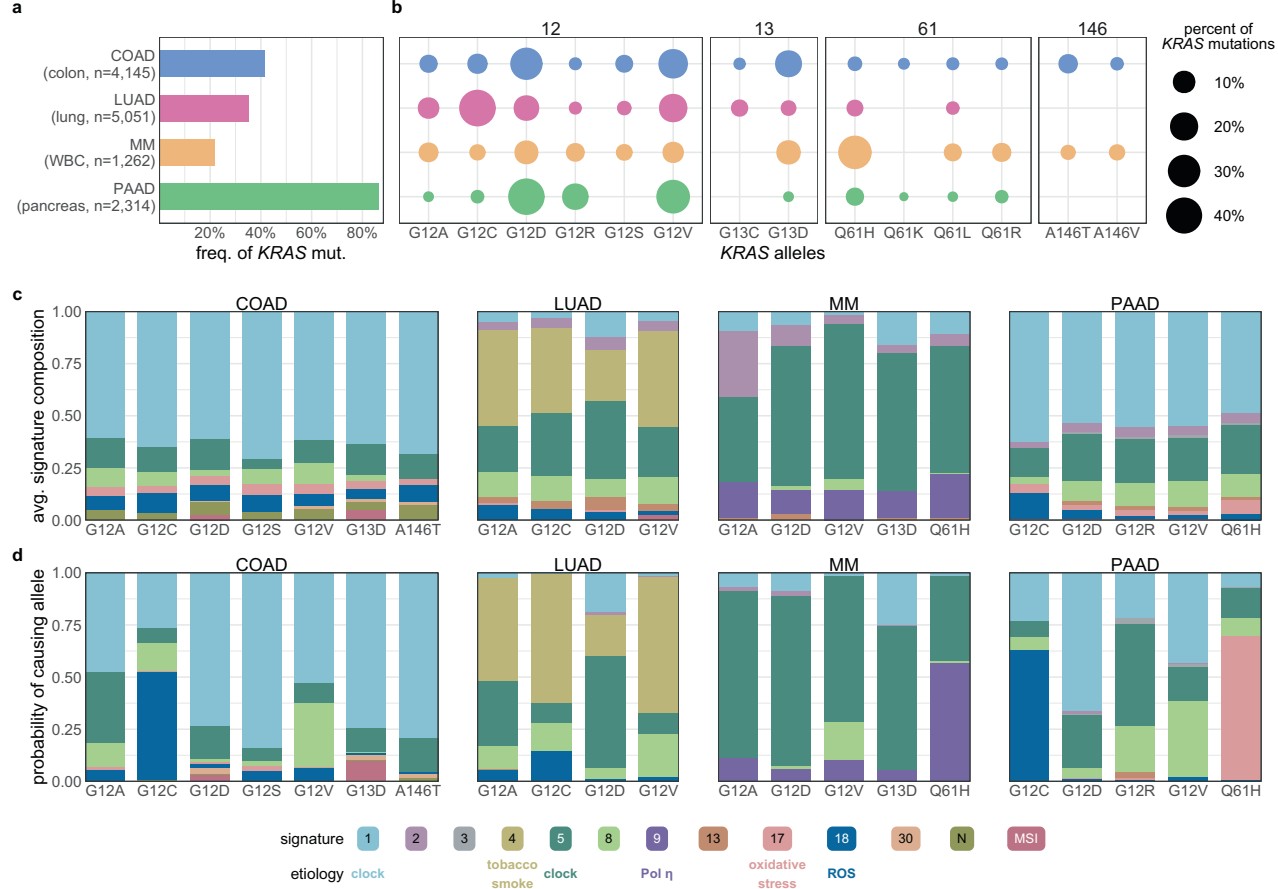

**Fig. 1 The contribution of mutational processes to *KRAS* mutagenesis. a** The frequency of *KRAS* mutations in each cancer. **b** The distribution of *KRAS* allele frequencies at the four hotspots, codons 12 (left), 13 (middle-left), 61 (middle-right), and 146 (right) in each cancer. The size of the circle reflects the percent of *KRAS* mutations that are the indicated allele in each cancer. Each cancer is assigned a different color. The number of tumor samples whose sequencing data was collected for this study is indicated along the y-axis. **c** The average composition of mutational signatures in tumor samples grouped by *KRAS* allele. Each color represents a different mutational signature. Mutational signatures of know etiology are annotated. **d** The average probability of each mutational signature to have caused the *KRAS* mutation in a tumor sample. This value accounts for the level of each mutational signature in the tumor sample, and the ability of the mutational signature to cause the indicated *KRAS* allele. In **c** and **d**, only *KRAS* alleles found in at least 15 tumor samples of the cancer type are included. Source data are provided in the Source data file.

frequency of *KRAS* was not caused primarily by distinct compositions of mutational processes in individual tumors.

Each mutational process has a different propensity to induce each *KRAS* allele. To discern if specific mutagenic processes were more likely to have caused particular *KRAS* alleles, the trinucleotide context of the *KRAS* mutation and the relative activity of the mutational signature in that tumor were used to calculate the probability that the allele in an individual tumor was caused by any detectable mutational process (Fig. 1d). In general, such probabilities reflected the underlying distribution of signatures, as seen in the similarities between Fig. 1c, d, suggesting that, while the mutational processes were capable of causing the observed *KRAS* mutations, they did not strictly determine which mutation was acquired.

In many cases, specific mutational signatures were much more likely to have caused the observed mutation than expected based on their background frequencies. For example, in COAD and PAAD, SBS18 (navy blue bars), likely caused by damage from reactive oxygen species[29,30], was strongly associated with G12C mutations (Fig. 1d and Supplementary Fig. 3a, d). This corroborated the previous finding that *KRAS* G12C mutations are more frequent in patients with MUTYH-associated polyposis[29], an autosomal recessive disease form of COAD

caused by biallelic loss-of-function mutations to the gene encoding the DNA glycosylase, *MUTYH*, responsible for clearing 8-oxoguanine:A mismatches that can cause the G12C mutation. In LUAD, the *KRAS* G12A/C/V mutations were primarily attributable to mutations caused by tobacco smoke, whereas *KRAS* G12D mutations were most likely attributable to clock-like mutations (Fig. 1d and Supplementary Fig. 3b). In MM, SBS9, associated with mutations introduced by polymerase η repair of activation-induced deaminase (AID) activity[26,31,32], was strongly linked with Q61H (Fig. 1c, d and Supplementary Figs. 2c and 3c), the most common *KRAS* mutation in that cancer. SBS8, of unknown etiology, had a substantial probability of causing several of the *KRAS* alleles, particularly G12V, across all four cancers (Fig. 1d, and Supplementary Figs. 2 and 3). SBS17, also of uncertain etiology though linked to oxidative stress in other cancers[33], was likely the primary cause for Q61H mutations in PAAD (Fig. 1d and Supplementary Fig. 3d).

**The frequency of most *KRAS* alleles cannot be solely attributed to the prevalence of detected mutagens.** The extent to which mutational signatures represent the mechanism driving *KRAS* allelic diversity was further analyzed by calculating the predicted

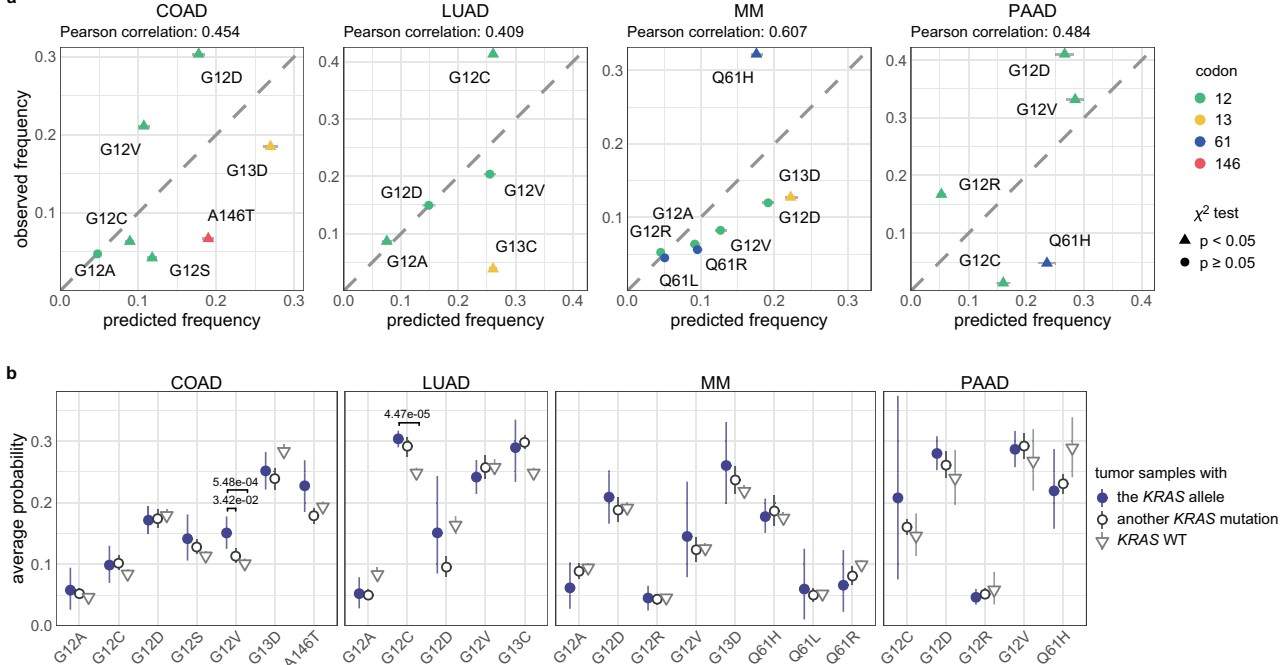

**Fig. 2 The predicted frequencies of cancer-specific *KRAS* alleles. a** The predicted versus observed frequency of *KRAS* alleles for the common alleles of each cancer. Triangles indicate rejection of the null hypothesis that the observed and predicted frequencies are the same ($\chi$-squared test, *p* values were adjusted using the Benjamini–Hochberg FDR correction method, hereon referred to as FDR-adjusted *p* values; FDR-adjusted *p* value < 0.05); circles indicate the failure to reject the null hypothesis ($\chi$-squared test, FDR-adjusted *p* value ≥ 0.05). Error bars indicate bootstrapped 95% confidence intervals of the predicted values. **b** The average probability of the indicated *KRAS* allele in tumors samples with the *KRAS* allele (closed circle), tumors samples with a different *KRAS* mutation (open circle), and tumor samples with WT *KRAS* (upside-down triangle). The errors bars indicate bootstrapped 95% confidence intervals of the mean. For each allele, differences in the probabilities between tumor samples with the allele and those with another allele, and between tumor samples with the allele and those with WT *KRAS* were tested via a Wilcoxon rank-sum test (FDR-adjusted *p* values < 0.05 are indicated). Source data are provided in the Source data file.

frequency of each allele based on the frequency of mutations in the same trinucleotide context throughout the exome or genome (Fig. 2a and Supplementary Data 6). The null hypothesis tested was that, assuming the cancer would acquire a *KRAS* mutation and one of the common alleles (found in >3% of the tumor samples for a given cancer) was sufficient, the frequency of the *KRAS* alleles would be determined by the mutational processes alone. The average predicted frequencies across the samples of each cancer were compared against the observed allele frequencies (Fig. 2a and Supplementary Data 6).

In COAD, G13D was predicted to be the most frequent allele (27%) but was observed less frequently (20%). The frequencies of G12S and A146T mutations were also overestimated, whereas G12D/V mutations were considerably underestimated. All are statistically significant and denoted by triangles in Fig. 2a ($\chi$-squared test, FDR-adjusted *p* < 0.05). In LUAD, the frequencies of the G12A/D/V alleles were accurately predicted, but the frequency of the most common allele, G12C, was substantially underestimated. The high frequency of this allele has been attributed to its association with SBS4 caused by tobacco smoke (Fig. 1c, d), but our observation suggests that there is additional biological pressure promoting this mutation in LUAD. The frequencies of the *KRAS* alleles were best predicted in MM, with an exception for the most frequent allele, Q61H, which was dramatically underestimated with a predicted frequency of 15.0%, but an actual frequency of 35.7% of *KRAS* mutations. In PAAD, all of the alleles were observed at a significantly different frequency than predicted by mutational signatures. In particular, the G12R mutation is expected to occur in 5.2% of PAAD tumors, which is far below the actual frequency of 16.7%. Overall, the

Pearson correlations between the observed and predicted *KRAS* allele frequencies for each cancer ranged from 0.4 to 0.6 (or 0.7–0.9 when restricted to just G12 alleles). Although the relatively high correlations, the significant discrepancy between observed and predicted frequencies suggests that the allelic distributions of *KRAS* were not solely determined by the prevalence of their respective causative single-nucleotide substitutions.

We also conducted a similar analysis considering those alleles that were left out in the previous analysis due to their low observed frequency in a given tumor type, but are frequent in another tumor type (Supplementary Fig. 4 and Supplementary Data 7). The alleles never or rarely found in a cancer were predicted to occur at frequencies ranging from 1.5% (for Q61L in PAAD) to 10.5% (for Q61K in LUAD), indicating that these alleles are not rare because their causative mutations do not occur, but instead because of weak oncogenic fitness in the tissue. For instance, *KRAS* A146T was predicted to be 8.9% of *KRAS* mutations in PAAD, but is exceedingly rare in this cancer, consistent with the previous demonstration that forced expression of *KRAS* A146T in mouse pancreas does not induce pancreatic intraepithelial neoplasia[4].

Another approach to examine the impact of mutagenic processes on allele specificity was to compare the probability of obtaining a certain *KRAS* mutation between tumor samples with the specific mutation, a different *KRAS* mutation, or WT *KRAS* (Fig. 2b). In most cases, tumors samples with a specific *KRAS* allele did not, on average, have a higher probability of obtaining that mutation than other tumors of the same cancer type. However, this was not true for *KRAS* G12V in COAD and *KRAS*

G12C in LUAD (Wilcoxon rank-sum test, FDR-adjusted *p* value < 0.05). Interestingly, the *KRAS* G12V mutation in COAD is likely to be caused by mutational signature SBS8 (Fig. 1d, and Supplementary Figs. 2a and 3a). The cause of this signature is currently unknown, though this result indicates that it plays an important role in *KRAS* G12V mutagenesis. The increased probability of a *KRAS* G12C mutation in tumor samples that did obtain the allele compared to *KRAS* WT LUAD tumor samples is likely due to the strong association between this mutation and signature SBS4 induced by carcinogens in tobacco smoke. However, no difference was detected between tumor samples with *KRAS* G12C and a different *KRAS* mutation, indicating that this mutagenic force is not specifically favoring the G12C allele. Overall, these results suggest that the probability of acquiring a particular *KRAS* allele was not significantly greater in tumor samples that did obtain the *KRAS* mutation.

Taken together, these results indicate that while the active mutational processes in a tissue contributed to which *KRAS* mutation was gained, they were not deterministic. Rather, how the unique biological properties of an allele interact with the preexisting signaling context of the tissue, often modified by additional mutational events, is likely a crucial factor in determining its frequency in cancer. This explanation for the distribution of *KRAS* alleles warranted further investigation into their genetic interactions.

**The *KRAS* alleles have distinct comutation networks**. We reasoned that if biological selection is driving *KRAS* allele selection in cancer, then distinct functions of each mutant form of KRAS would be reflected in cooperating genetic events. An increased frequency of comutation with another gene suggests a cooperative effect, whereas a reduced frequency of comutation (compared to random) suggests that the second event is functionally redundant or that it introduces an inhibitory effect. The extreme of the latter effect is commonly known as "mutual exclusivity." For instance, in COAD, *APC* comutation enhances the effects of oncogenic *KRAS*-induced hyperactivation of the Wnt signaling pathway, essential for the growth of cancer stem cells in the intestinal crypts[34]. Alternatively, in LUAD, the mutational activation of *EGFR* was demonstrated to be cytotoxic in the presence of a *KRAS* mutant, and, thus, the two are rarely found in the same tumor[35,36].

The comutation interactions between each *KRAS* allele and every other mutated gene were investigated using a one-sided Fisher's exact test of association to identify increased rates of comutation and a test for mutual exclusivity proposed by Leiserson et al.[37] to identify reduced rates of comutation (Supplementary Data 8). To reduce the number of false positive interactions, multiple filters were applied to restrict which genes were tested, including only testing for increased or reduced comutation interactions with genes mutated in at least 1% or 2% of tumor samples of a cancer type, respectively (see "Methods"). The result of the comutation analysis on COAD tumors was a weakly connected network of the *KRAS* alleles with only a few genes linking the alleles together (Fig. 3a). These linking genes tended to be well-studied cancer genes, such as *BRAF* (primarily V600E mutations), *APC* (mostly nonsense truncating mutations), and *TP53* (primarily mutations in the sequence encoding the DNA-binding domain of the protein). Contrary to a common assumption, while *KRAS* and *TP53* were frequently found mutated in the same tumor, there was a detectable reduction in comutation between *TP53* with *KRAS* G12D and G13D compared to the rest of the alleles (Fig. 3b).

Consistent with the idea that each allele is functionally distinct, a substantial number of genes comuted with just one *KRAS*

allele. To gain functional insight into the network, genes known to physically interact with KRAS[16], signal upstream or downstream of KRAS[38], or are known oncogenes or tumor suppressor genes[39] were extracted (Fig. 3b). Several *KRAS* alleles had reduced comutation with *NRAS* and *BRAF*, and increased comutation with *APC* and *PIK3CA*, interactions that have been previously documented[34,40–49]. Similar to *KRAS*, *PIK3CA* mutations tend to occur in several hotspots, each likely having slightly different effects on hyperactivation of the protein. However, specifically testing for comutation between *KRAS* alleles, and the most common *PIK3CA* mutations did not reveal any strong preferences for particular activating *PIK3CA* mutations.

Some novel interactions included increased comutation of *PORCN* with *KRAS* A146T, *MTOR* with G12C, and *SMAD4* with G12V. *KRAS* G12V had an increased rate of comutation with *TCF7L2*, which encodes TCF4, a regulator of Wnt signaling often dysregulated in COAD[49–51], specifically the R488C mutation. Further, several of the alleles showed enrichment for cellular functions in their comutation networks (Fig. 3c). One of the strongest effects was an enrichment in the G12D comutation network of interactors with YWHAZ, a 14–3–3 scaffolding protein implicated in modulating many interactions, including the activity of Rho GEF 7 on RAC1 in phagocytosis and cell adhesion[52]. Also, genes involved in the Hippo and Wnt signaling, key pathways in COAD, were enriched in the comutation networks of *KRAS* G12V. The comutation network of the G13D allele was enriched for genes implicated in apoptosis and senescence. Additional genes of interest that had comutation interactions with *KRAS* G12D are shown in Fig. 3d, e. These include increased comutation with *AMER1*, a negative regulator of Wnt signaling[53,54].

The *KRAS* allele-specific comutation network uncovered in LUAD was far larger than that of COAD (Supplementary Fig. 5a). This was likely caused by the higher mutation frequency in this cancer, increasing the statistical power to detect both increased and reduced comutation interactions. As in the network derived from COAD, many of these genes were involved in integral KRAS signaling pathways, including an increased comutation interaction between *KRAS* G12A and *MAP2K3*, a reduced comutation interaction between *KRAS* G12D and *ERBB4*, and a very strong increased rate of comutation between *KRAS* G12C and *STK11* (Supplementary Fig. 5b). There were several intriguing cellular processes enriched in the LUAD networks for each allele (Fig. 3c). For example, *KRAS* G12C had comutation interactions with many genes encoding proteins that interact with Myc ("PPI of MYC (TF)"), and the G12D comutation network was enriched for interactions with focal adhesion genes.

Conducting this analysis in MM was hampered by the fact that this cancer is known to be frequently multiclonal[55,56]. As such, some detectable comutation events were mutations acquired by distinct populations in a single patient, potentially obfuscating true comutation interactions. Due to this caveat, limiting the analysis to genes known to be recurrently mutated in MM reduced the chance of highlighting a false positive[55]. From this limited scope, it was discovered that *NRAS* had reduced comutation with *KRAS* G12D, Q61L, and Q61R, but one of the highest rates of comutation (18.5%) with *KRAS* Q61H, the most common *KRAS* mutation in MM (Supplementary Fig. 6). Interestingly, this was just below the rate of *NRAS* mutation in *KRAS* WT tumors (23.6%), suggesting that the signaling of the Q61H allele is fundamentally different from the other *KRAS* mutations in MM, especially G12D. Of these comutation events, the *NRAS* mutations were mostly at codon 61, common for *NRAS*-driven cancers, such as skin cutaneous melanoma[57,58], and there was no detectable pattern of comutation between particular *KRAS* and *NRAS* alleles.

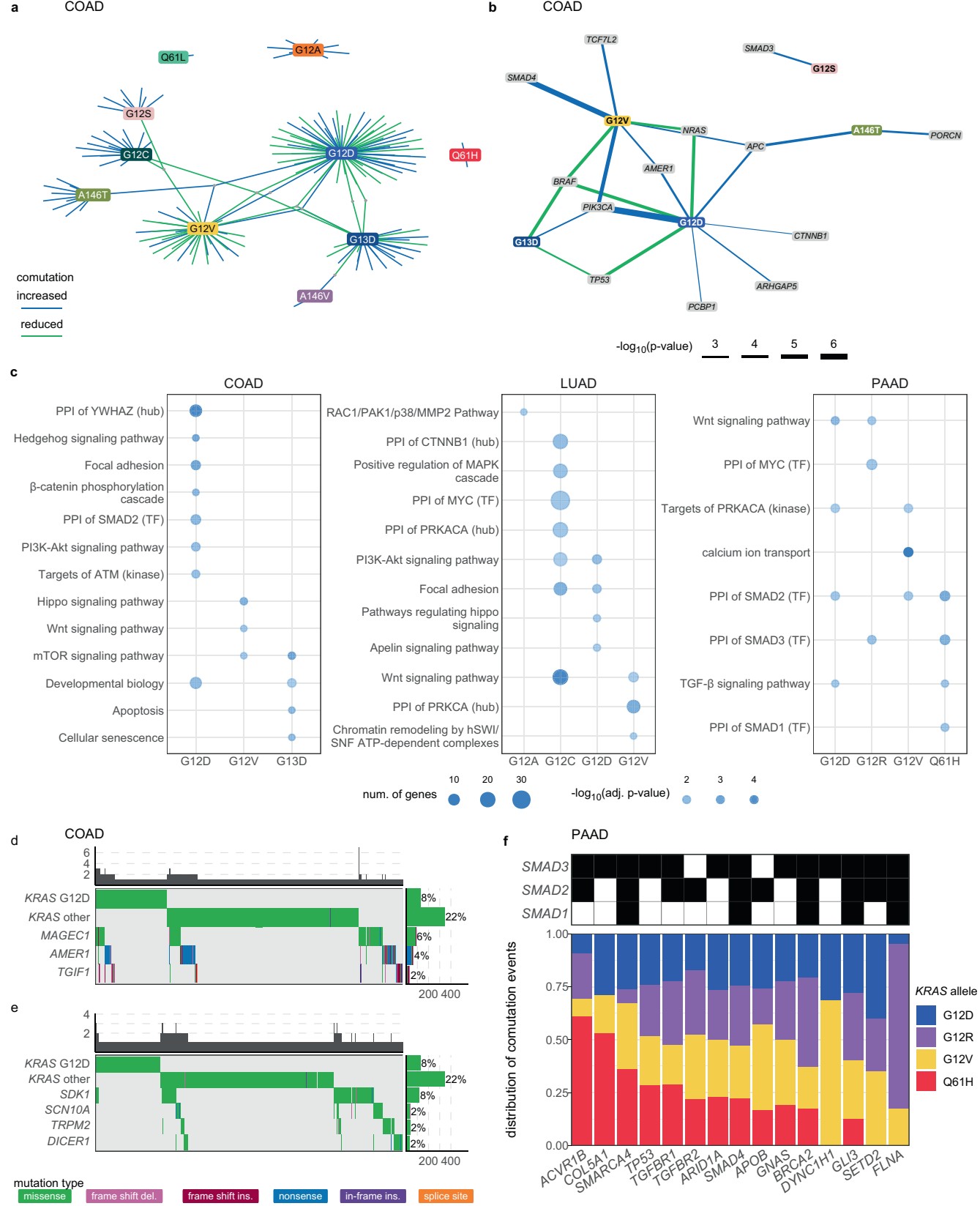

**Fig. 3 The comutation networks of oncogenic *KRAS* alleles. a** The comutation network of the *KRAS* alleles in COAD with each edge representing a significant comutation interaction between an allele and another gene (*p* value < 0.01). The color of the edge indicates whether the interaction was an increase (blue) or decrease (green) in the frequency of comutation. Genes with multiple interactions are represented by a gray dot to disambiguate them from where edges intersect. **b** A subset of the network shown in **a** of genes that encode proteins known to physically interact with KRAS, are in one of its canonical upstream or downstream pathways, or are validated oncogenes or tumor suppressors. The width of the edge indicates the strength of the association. **c** Cellular functions enriched in the comutation networks of the *KRAS* alleles in COAD (left), LUAD (center), and PAAD (right). The size of the dot indicates the number of genes in both the function and the comutation network, and the transparency indicates the FDR-adjusted *p* value of the enrichment. **d, e** A visualization of the increased (**d**) or decreased (**e**) comutation of select genes with *KRAS* G12D in COAD. Rows of the central plot represent genes. Each column of the central plot is a different tumor sample. A filled space denotes a mutation of the gene in the sample, the color describing the type of variant. The bar plots above and to the right indicate the marginal values of the central plot. **f** A comparison of the comutation frequencies in PAAD of the genes producing proteins in the PPIN of SMAD1-3. Each column is a gene with a comutation interaction with a *KRAS* allele and in at least one of the gene sets. The black tiles on top indicate that the gene was in the PPIN of the indicated SMAD protein. The bar plot shows the distribution of the comutation events of each gene across tumor samples with the various *KRAS* mutations. *n* = 4145 COAD, 5051 LUAD, 1262 MM, and 2314 PAAD biologically independent tumor samples for the increased comutation analysis, and *n* = 1536 COAD, 891 LUAD, 1395 PAAD biologically independent tumor samples for the reduced comutation analysis. Source data are provided in the Source data file.

The *KRAS* allele comutation network found in the PAAD tumor samples demonstrated that many genes had detectable comutation interactions with multiple alleles, primarily of reduced comutation (Supplementary Fig. 7a). There were numerous genes that had opposing comutation interactions with different alleles. Of these, four interact with or signal through KRAS[16,38] or are known oncogenes or tumor suppressors[39]: *TP53*, *RNF43*, *MAP2K4*, and *RBM10* (Supplementary Fig. 7b, c). Notably, while *TP53* tended to comutate with *KRAS* G12V, it was at a significantly lower rate than expected by random chance, given the overall mutation rate of *TP53* and the mutational burden of the tumors. *TP53* was primarily mutated at known hotspots R175, R248, R273, and R282 (refs. [59–61]), or had nonsense or frameshift mutations. Most of the mutations to *RNF43* and *RBM10* were nonsense or frameshift mutations. *MAP2K4* primarily had missense mutations at known mutational hotspots[61].

There were many notable cellular functions and processes enriched in the comutation networks of the *KRAS* alleles (Fig. 3c), including the protein–protein interaction networks (PPIN) of SMAD1-3 and TGF-β signaling. While these SMAD gene sets were related, the underlying comutation interactions that drove the enrichment were different for each *KRAS* allele (Fig. 3f). For instance, the comutation events of *ACVR1B* with *KRAS* were primarily with Q61H, whereas those with *FLNA* were mostly with G12R. These subtle differences suggest that specific and nuanced alterations of SMAD signaling best complement a given *KRAS* allele in PAAD.

It is important to note that many of the comutation interactions identified from this allele-specific analysis were not identified from a gene-level analysis that disregards the *KRAS* allele information (Supplementary Data 9). For instance, the number of genetic interactions with reduced comutation in the non-allele-specific analysis was 105 for colon, whereas the number in the allele-specific analysis was 63. Among these, only 35 were in common (Supplementary Fig. 8a). The overlap for increased comutation and other tumor types are similarly small (Supplementary Fig. 8), underscoring the importance of the allele-specific analysis.

**KRAS allele-specific genetic dependencies reveal potential synthetic lethal vulnerabilities.** The perturbations necessary to drive cancer expose vulnerabilities that are not present in the normal cell-of-origin. For example, the MSI that often leads to cancer simultaneously makes the inhibition of Werner syndrome ATP-dependent helicase (WRN) lethal to the tumor cells[62,63]. As the *KRAS* alleles have measurably different signaling behaviors

and genetic interactions, they likely have specific genetic vulnerabilities. To this end, we used data from a genome-wide, CRISPR/Cas9 knockout screen of cancer cell lines[64,65] to identify genes with *KRAS* allele-specific genetic dependencies. The analysis was restricted to *KRAS* alleles for which there were at least three different cell lines with the mutation, limiting the following investigation to only COAD and PAAD cell lines. Allele-specific enrichments for signaling pathways and cellular processes were identified using gene set enrichment analysis (GSEA)[66], and individual genes demonstrating differential genetic dependency by *KRAS* allele were identified using ANOVA (*p* value < 0.01) and *t* tests (FDR-adjusted *p* value < 0.05).

For COAD, there was a sufficient number of cell lines with WT *KRAS* or G12D, G12V, and G13D mutations for this analysis. Measuring for gene set enrichment revealed strong patterns in differential dependency of various cellular processes (Fig. 4a). For example, genes involved in ERBB4 signaling tended to have a weaker lethal effect when knocked out in cell lines with *KRAS* G12V mutations than in *KRAS* G12D, G13D, or WT cell lines (Fig. 4b). Similarly, the *KRAS* G13D cell lines were less affected when genes involved in oxidative phosphorylation were targeted (Fig. 4c). To discover individual genes with allele-specific interactions, each gene was tested for differential genetic dependency with the cell lines grouped by their *KRAS* allele. The resulting 62 genes were hierarchically clustered into four groups by their dependency scores (Fig. 4d and Supplementary Data 10). Genes in cluster 2 tended to have stronger genetic dependency in cell lines with *KRAS* G12V, while those in cluster 3 demonstrated weaker dependency in G12D cell lines. Four notable genes with allele-specific associations are displayed in Fig. 4e. First, knocking out *LIN7C*, a gene that maintains the asymmetric distribution of membrane proteins in polarized epithelial cells[67], had a more severe reduction on growth in *KRAS* G13D cell lines compared to the others (Fig. 4e). Also, a regulator of apoptosis previously linked to dysregulated expression in cancer[68], *TFPT*, demonstrated significantly greater dependency in G12D cell lines. Interestingly, *STARD9*, a gene encoding a kinesin required for mitotic spindle assembly[69], had moderate growth defects when knocked out in all cell lines except those with a *KRAS* G12D mutation. Lastly, the kinetochore-associated protein (*KNTC1*), a regulator of the mitotic checkpoint[70,71], which demonstrated moderate to strong lethal effects when knocked out in almost every cell line except for those with a *KRAS* G12V allele (Fig. 4e).

For the genetic dependency analysis of PAAD, the *KRAS* alleles with a sufficient number of cell lines were G12D, G12R, and G12V (there were not enough WT *KRAS* cell lines to include in the analysis). GSEA revealed substantial differences in the

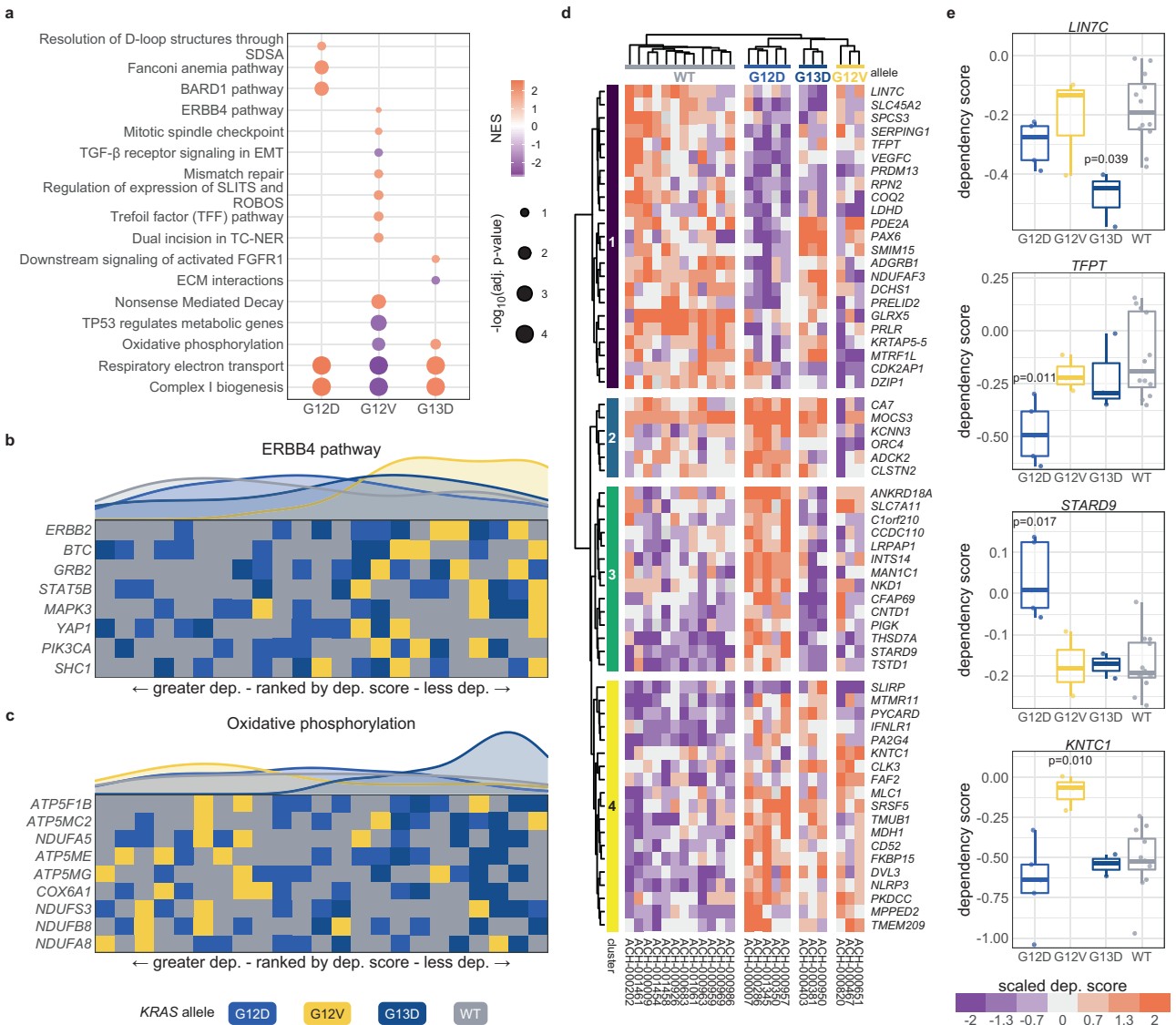

**Fig. 4 Allele-specific genetic dependencies in COAD cell lines. a** Gene sets with significant enrichment for increased (lower dependency score; purple) or reduced (higher dependency score; orange) genetic dependency in COAD cell lines. The size of the dot relates the FDR-adjusted *p* value of the association and the color indicates the strength of the enrichment ("normalized enrichment score"). **b, c** Heatmaps ranking the cell lines by dependency ("dep.") score of the genes at the leading edge of enrichment for two gene sets. Each row represents a gene and each cell represents a cell line colored by its *KRAS* allele. The cell lines are arranged in ranking order by their dependency score for the gene. Thus, each column indicates a rank. The line plots above the heatmaps indicate the representation (density) of each *KRAS* allele at each rank across the genes. **d** Hierarchically clustered heatmaps of the genes that demonstrated differential genetic dependency amongst cell lines of different *KRAS* alleles. Each column is a cell line labeled by its DepMap identifier and each row is a gene. **e** Examples of genes that demonstrated differential genetic dependency amongst cell lines of different *KRAS* alleles (*t* tests; FDR-adjusted *p* values). For the box plots, the box demarcations represent the 25th, 50th, and 75th percentiles, and the whiskers extend from the box to the largest and smallest data points at most 1.5 times the interquartile range away from the median. *n* = 23 biologically independent COAD cell lines. Source data are provided in the Source data file.

dependencies of critical cellular pathways (Supplementary Fig. 9a). For instance, the G12D cell lines demonstrated a reduced dependency on the genes at the G2 and M DNA damage checkpoint (Supplementary Fig. 9b). Moreover, the G12R cell lines were less dependent on PI3K signaling downstream of FGFR1, driven through a reduced dependency on *FRS2* (fibroblast growth factor receptor substrate 2) and *GRB2*, which encodes a protein linking EGFR to the GEF SOS1 (Supplementary Fig. 9c). Similarly, the cell lines with *KRAS* G12V mutations were less sensitive to the knockout of genes implicated in cellular senescence (Supplementary Fig. 9d). This enrichment was driven by a significantly reduced dependence upon *JUN*, which encodes

the transcription factor c-JUN, and a beneficial impact on growth (a positive dependency score) from knocking out *MAPK8* (JNK-1), which regulates c-JUN via phosphorylation (Supplementary Fig. 10). In these cell lines, 130 individual genes demonstrated *KRAS* allele-specific genetic dependency (Supplementary Fig. 10a and Supplementary Data 11). Several noteworthy interactions include a regulator of cell cycle progression, *KHDRBS1* (ref. [72]), the oxygen sensor, *EGLN2* (ref. [73]), and a stabilizer of p53, *BRI3BP*[74] (Supplementary Fig. 10b). Overall, the *KRAS* alleles were associated with substantially different genetic dependencies on specific cellular processes, signaling pathways, and individual genes.

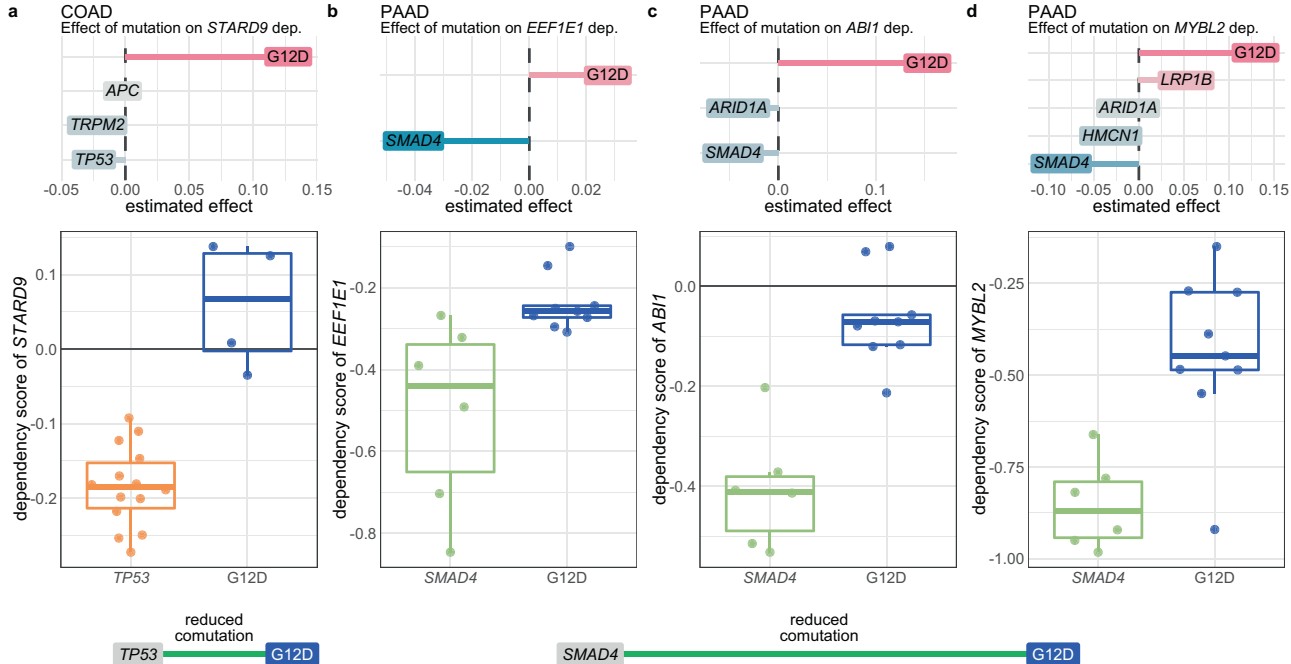

**Fig. 5 Some dependency interactions can be explained by comutation events. a** The nonzero coefficients for the model of *STARD9* dependency in COAD cell lines regressed on *KRAS* G12D (versus all other *KRAS* alleles) and its comutation interactors (top), and the actual dependency scores for *KRAS* G12D mutant and *TP53*-mutant cell lines (bottom). Cell lines without either mutation or with both are not shown. *KRAS* G12D has reduced comutation with *TP53* in COAD. **b–d** The nonzero coefficients for the models of **b** *EEF1E1*, **c** *ABL1*, and **d** *MYBL2* dependency in PAAD cell lines regressed on *KRAS* G12D (versus all other *KRAS* alleles) and its comutation interactors (top), and the actual dependency scores for *KRAS* G12D mutant and *SMAD4* mutant cell lines (bottom). Cell lines without either mutation or with both are not shown. *KRAS* G12D has reduced comutation with *SMAD4* in PAAD. For the box plots, the box demarcations represent the 25th, 50th, and 75th percentiles, and the whiskers extend from the box to the largest and smallest data points at most 1.5 times the interquartile range away from the median. *n* = 23 biologically independent COAD cell lines, and *n* = 23 biologically independent PAAD cell lines. Source data are provided in the Source data file.

**An integrated analysis of allele-specific comutation and genetic dependencies.** As emphasized by the weakly connected comutation networks, the *KRAS* alleles are not acting in the same genetic environments, and, therefore, their allele-specific genetic dependencies might be mediated by a comutating partner. To address this hypothesis, we constructed linear models for the dependency score of each gene with allele-specific dependency that included a coefficient for the previously linked *KRAS* allele and a coefficient for the mutation of each gene in its comutation network. These models were then fit with elastic net regression to isolate the most informative predictors, adjusting for the RNA expression of the targeted gene[75].

Some of the models indicated that the mutation of a comutation partner could explain the allele-specific dependency interaction. An example of this was how the dependency of COAD cell lines on *STARD9* was greater in *TP53*-mutant lines than in *KRAS* G12D lines (Fig. 5a). Most of the *TP53* mutations were located in the DNA-binding domain, two of which were nonsense mutations. Of the other mutations, two were at splice sites, one was in the nuclear localization signaling domain, and two more were either nonsense or frameshift mutations in the N-terminal domain. All were either predicted to be deleterious[76,77] or previously identified recurrent mutations[61]. If *TP53* mutations induce a stronger dependency on *STARD9*, the reduced frequency of comutation between *TP53* and *KRAS* G12D would cause the opposite effect to be ascribed to the G12D allele. A similar effect was found between *KRAS* G12D and *SMAD4* in PAAD cell lines for the dependency on *EEF1E1*, *ABL1*, and *MYBL2* (Fig. 5b–d). All but two of the *SMAD4* mutations were frameshift or nonsense

mutations. Because of the reduced comutation interaction between *KRAS* G12D and *SMAD4* in PAAD, the effects of knocking out these genes can be ascribed to an allele-specific effect or to the *SMAD4* mutation. These examples highlight how the allele-specific comutation interactions of *KRAS* can influence the interpretation of other interactions.

## Discussion

This study addresses the genetic complexity of cancer through a comprehensive genetic interaction analysis of oncogenic *KRAS* alleles in COAD, LUAD, MM, and PAAD. Measuring the levels of mutational signatures revealed that the cancer-specific distributions of *KRAS* mutations were influenced, but not determined, by the active mutational processes in the tumor samples. This result suggests that the biological properties of the *KRAS* alleles, within the context of the tissue-of-origin, is an important factor in the positive selection of a *KRAS* mutation during the evolution of a tumor. Indeed, we have previously demonstrated that mutant forms of KRAS produce distinct molecular and cellular phenotypes that are largely dependent upon the tissue context[4,78,79]. To investigate allele-specific genetic properties, we conducted statistical tests to identify patterns of comutating genes and genetic dependencies for each *KRAS* allele in each cancer. The former identified genes that comutated with specific *KRAS* alleles at an unexpectedly high frequency, suggesting that they were alterations that cooperated with the *KRAS* allele to promote tumor growth. Alternatively, some genes comutated with a *KRAS* allele less frequently than expected by chance, suggesting they

were functionally redundant mutations or introduced an inhibitory effect on the tumor's progression. Finally, functional interactions were identified between *KRAS* alleles and cellular processes and individual genes. Together, these findings support a model in which the various oncogenic *KRAS* mutations are not biologically redundant, but instead have distinct properties that are reflected in their genetic interactions.

This analysis of *KRAS* genetic networks in four different tumor types highlights the tissue-specific nature of genetic interactions. In places, we focused on the results from the analysis of COAD, as it demonstrated a high variability in the types of *KRAS* alleles, had limited exogenous mutational pressure (in contrast to the effects of smoking-induced mutations in LUAD), and we had a large number of WGS and whole-exome sequencing (WES) data. However, allele-specific genetic interactions were not consistent between tissues, demonstrating the complex relationship between the tissue-of-origin, KRAS function, and cooperating genetic events. While the intrinsic biochemical properties of a KRAS mutant are likely maintained in each cancer, their downstream signaling properties, and ultimately their effects on tumorigenesis, are determined by the basal configuration of the tissue-specific signaling network[78]. Thus, the configuration of the tissue signaling network influences the genetic interactions that arise during cancer progression.

In addition to the importance of tissue specificity, this study provides compelling evidence that the somatic missense mutations that activate oncogenes are not always equivalent. We and others have demonstrated the distinct effects of the *KRAS* alleles, both computationally and experimentally, revealing many instances of substantial variation between different mutations of the same gene. This is likely a more general principle applicable to many oncogenes, especially those with multiple mutational hotspots.

The *KRAS* allele-specific comutation analysis indicates that the various *KRAS* mutations act within distinct genetic environments. This likely impacts the effects of therapeutics, potentially obfuscating the underlying reason for disparate responses in clinical trials. The principle of this phenomenon was demonstrated by the analysis of the CRISPR-Cas9 screen, when the comutation events were included as explanatory covariates: in several instances, the allele-specific dependency originally assigned to a specific *KRAS* mutation could instead be attributed to an allele-specific co-mutant gene. Thus, we provide evidence that not only do the biological properties of the *KRAS* alleles contribute to their effect on the tumor, but so too do their unique genetic interactions.

Finally, this study has broad implications for the understanding of oncogene biology and for cancer therapy. Whether a targeted therapy directly inhibits the activated oncoprotein or not, it is important to understand how allele-specific signaling properties and genetic interactions influence therapeutic response. For instance, *BRAF* activating mutations have been classified into three groups defined by their functional effects on the protein product[80,81], which consequently determines their response to different inhibitors[82,83]. Moreover, the response of HER2 mutant cancers to HER2 inhibition varies depending on the tissue-of-origin of the cancer[84], which could be due to intrinsic signaling differences between the tissues-of-origin or to cooperating mutations unique to a specific cancer type. For cancer therapy to be truly precise, it will be key to appreciate and understand the complexity of the genetic networks in each cancer type.

## Methods

**Cancer sample data sources and acquisition**. WGS, WES, and targeted gene panel sequencing ("targeted-sequencing") data were collected of COAD, LUAD, MM, and PAAD. WES and WGS data were downloaded from cBioportal[85,86], which included relevant projects from The Cancer Genome Atlas (TCGA)[49,87,88]

and other smaller studies. Additional data were acquired from the International Cancer Genome Consortium (ICGC) for pancreatic cancer and colorectal cancer[89]. MM WES data were gathered from the Multiple Myeloma Research Foundation (MMRF)-CoMMpass online repository[90]. Panel data for multiple cancers were retrieved from AACR Project Genomics Evidence Neoplasia Information Exchange (GENIE v5)[91]. GENIE data are an aggregation of several different panels ranging from 30 to 600 genes. *KRAS* was included in all of the libraries. A detailed list of all cancer studies can be found in Supplementary Data 1 and 2, and links to access the data are provided in the "Data availability" section of the "Methods."

**Hypermutated sample cutoff**. Some of the COAD samples had five to ten times more mutations than the average, often due to MSI. A Gaussian mixed model was used to find the optimal cutoff based on available WGS and WES data. The top 17% and 21% of samples were considered hypermutants in WGS and WES, respectively. The same 17% cutoff was applied to the targeted-sequencing data. Hypermutants were not excluded from the identification of mutational signatures because signature 6 (marked as "MSI") is caused by MSI.

**Tissue gene expression filter**. A conservative filter for tissue-specific gene expression was used to remove genes not expressed in the tissues of study. Normal tissue gene expression data were gathered from the GTEx Portal[92] (03 December 2018) and The Human Protein Atlas (HPA, 03 December 2018)[93], and tumor expression data were collected from MMRF-CoMMpass (14 January 2019), TCGA-COAD, TCGA-LUAD, and TCGA-PAAD[49,87,88,90]. A gene was considered "expressed" in a tissue if it met at least one of the following criteria: (1) a median expression level of at least one TPM across all samples of the tissue in GTEx, (2) indicated as expressed at at least one TPM in the HPA dataset for the tissue, (3) expressed with a median level of 1 batch-normalized raw counts (using RSEM) in the corresponding tumor RNA-sequencing data.

**Calculating overall distribution of hotspot mutations**. The frequency of mutations at the four hotspots of *KRAS* across COAD, LUAD, MM, and PAAD was calculated by accounting for the different yearly incidence of each cancer type. The incidence of cancers of the "colorectum," "lung and bronchus," "myeloma," and "pancreas" were obtained from the American Cancer Society[94]: 3,870,000 colorectum, 5,930,000 lung and bronchus, 680,000 myeloma, and 1,280,000 pancreas. The incidences of COAD, LUAD, and PAAD were estimated by multiplying the number of cases of their respective tissue by the proportion they constitute: 95%, 50%, and 95%, respectively[94,95]. The distribution of mutations to the hotspots across all cancers was calculated by finding the frequency within each cancer type, then combining those figures, weighting by their yearly incidence.

**Identifying mutational signatures**. The genome-wide mutations of a sample can be deconvolved into mutational signatures that represent endogenous or exogenous mutagenic processes[26]. Single-nucleotide variants from exomes or genomes were divided into 96 types, according to the six mutations of a pyrimidine (C > A, C > G, C > T and T > A, T > C, T > G) and the 16 possible combinations of 3′ and 5′ adjacent bases. The MATLAB[96] implementation of NMF algorithm, SigProfiler[26], was used to discover the underlying mutational patterns that are common across tumors. Mutational signatures were discovered separately for each tumor type, and the optimal number of signatures was determined based on silhouette width and Frobenius error[97].

The spectrum of the signatures discovered by NMF were matched to those documented by the Catalog Of Somatic Mutations In Cancer (COSMIC)[61]. For the signatures for which none of the 30 signatures in the COSMIC catalog was found to be compatible, we referred to more recently published studies and expanded upon the COSMIC catalog. In particular, there were multiple subtypes of signature 7 reported previously by Hayward et al. and Alexandrov et al.[27,98]. Further, the analysis revealed a signature that was predominantly C > A, but not a subtype of signature 7. This signature 38 was previously reported to be caused by indirect UV exposure[27]. Three versions of the signature associated with POLE mutations, signature 10, were discovered (previously reported by Alexandrov et al.[27]). These three POLE signatures differed in the C > A, C > T, or C > G parts of the mutational spectrum. In LUAD, a signature with mutations of type C[C > A]N and T[C > A]N attributable to 8-oxo-guanine[27] was found. One signature that was discovered in COAD did not have a good match with a previously published signature, although it resembled a signature previously reported to be caused by SBSA[99], and signatures 34 and 41 reported by Alexandrov et al.[27]. This signature was not adjusted to resemble those previously reported because the results from different studies were not in strong agreement. This signature, referred to as "N," did not contribute to *KRAS* mutations. Three of the signatures discovered via NMF were likely to be artifacts[100] and were removed from downstream analysis. Signatures that contributed to <5% of the mutations were also removed from downstream analysis. The levels of each signature in each tumor sample were calculated using nonnegative least squares and was restricted to signatures previously associated with the cancer type (as this reduces false assignment of signatures)[100]. The final spectra for each mutational signature and mutational signature composition of each tumor sample can be found in the Supplementary Data.

The levels of a particular mutational signature were compared between two groups of tumor samples separated by their observed *KRAS* allele using a Wilcoxon rank-sum test. The *p* values were adjusted for multiple hypothesis testing using the Bonferroni method.

**Probability of *KRAS* mutations from mutational signatures**. For each sample harboring a *KRAS* allele, the probability of each mutational signature to have caused the mutation was calculated by considering the weight of the base change among the 96 possibilities and the relative contribution of the signature to the mutations in the sample. Thus, the probability *p* of a tumor sample *a* to have acquired the *KRAS* mutation *k* from signature *s* of all signatures *S* can be calculated using Eq. 1.

$$p_{k,s,a} = \frac{c_{s,a} w_{k,s}}{\sum_i^S c_{i,a} w_{k,i}} \tag{1}$$

where $c_{s,a}$ is the contribution of signature *s* in sample *a*, and $w_{k,s}$ is the weight of mutation *k* in signature *s*. The probability is normalized to sum to 1 by dividing by the probability of getting the observed *KRAS* mutation from any of the signatures. The probability of a mutational signature to have caused a *KRAS* mutation was compared between two groups of tumor samples separated by their observed *KRAS* allele using a Wilcoxon rank-sum test. The *p* values were adjusted for multiple hypothesis testing using the Bonferroni method.

**Calculating the probabilities of *KRAS* alleles**. The mutational signatures are linear combinations of the 96-dimension spectrum of possible mutations (see "Identifying mutational signatures" above). Thus, assuming the null hypothesis that the prevalence of active mutational processes alone determines the frequency of *KRAS* alleles in a cancer, and the processes are active with the same probability throughout the genome, the probability of a tumor sample to acquire a specific *KRAS* allele was calculated as the frequency of the same mutation across the entire genome. For each cancer, the pool of possible *KRAS* mutations were restricted to those found in at least 3% of the tumor samples for the results presented in Fig. 2a, b, and those found in at least 3% of any cancer for the results presented in Supplementary Fig. 4. The average probability of each *KRAS* allele is presented in Fig. 2b with bootstrapped 95% confidence intervals around the mean using the "boot" R package and the "percentile" method[101,102]. A Wilcoxon rank-sum test was used to compare the distributions of the probabilities between tumor samples with the indicated *KRAS* allele and either tumor samples with a different *KRAS* mutation or *KRAS* WT tumor samples. The *p* values were adjusted for multiple hypothesis testing using the Benjamini–Hochberg FDR method.

**Predicting *KRAS* allele frequency**. The expected frequencies of the *KRAS* alleles were calculated as the mean probability of obtaining the *KRAS* allele across all tumor samples of a cancer type (see "Calculating the probabilities of *KRAS* alleles" above). The 95% confidence intervals around the mean were bootstrapped using the "boot" R package and the "percentile" method[101,102]. The predicted frequencies of the *KRAS* alleles for each cancer are available in Supplementary Data 6 and 7. A *χ*-squared tested was used to test the null hypothesis that there is no difference between the predicted and observed frequency for each *KRAS* allele. The *p* values were adjusted for multiple hypothesis testing using the Benjamini–Hochberg method (referred to as FDR-adjusted *p* values).

**Comutation with *KRAS* alleles**. A one-tailed Fisher's exact test of independence was used to identify increased frequency of comutation between *KRAS* alleles and other mutated genes. Only genes with an overall mutation frequency of at least 1% in the given cancer were considered. In addition, only comutation partners with at least three comutation events or a comutation frequency with a *KRAS* allele of at least 10% (i.e., 10% of the tumors with a *KRAS* allele also had a mutation in the given gene) were considered. Increased comutation interactions with a *p* value < 0.01 were considered statistically significant.

The row-column exclusivity test was used to identify reduced frequency of comutation between *KRAS* alleles and other mutated genes[37]. This is a permutation-based test that finds the probability of observing the actual number of mutually exclusive events given that the number of times the gene is mutated in all samples is fixed and the number of mutations in each sample is fixed. Thus, the test conditions on both the frequency of mutation of the gene and the mutational burden of the samples. For this reason, only WGS and WES data could be used for this analysis (using just the exonic mutations from WGS). Only genes with a mutational frequency of at least 2% and at least ten mutually exclusive events were considered. Reduced comutation interactions with a *p* value < 0.01 were considered statistically significant.

To further reduce the number of false positive comutation interactions reported between the *KRAS* alleles and genes previously reported to be involved in cancer, those that signal through KRAS, and genes that directly interact with KRAS, these sets of interactions were further filtered to fall below an FDR of 0.25 that is estimated using the Benjamini–Hochberg method. Only interactions that met this criterion are presented in Fig. 3b, and Supplementary Figs. 6b and 8b.

The Fisher's exact test was used to detect increased comutation interactions because, unlike the row-column exclusivity test, it could utilize the targeted-sequencing data. However, the row-column exclusivity test outperformed the row exclusivity test, a comparable permutation-based approximation of the Fisher's exact test, in the original publication by Leiserson et al.[37], suggesting it would be more sensitive for detecting reduced comutation interactions in the current study.

COAD samples identified as hypermutants were excluded from this analysis as they were likely microsatellite instable. Thus, these samples would be expected to have a high proportion of passenger mutations that would contribute substantial noise to the identification *KRAS* allele-specific comutation interactions.

**Functional enrichment**. The R interface to the online *Enrichr* tool was used to identify enriched gene sets in the comutation networks and allele-specific synthetic lethal clusters[103,104]. The online API was last accessed on April 9, 2020. Gene sets from the following sources provided by Enrichr were used: BioCarta (2016), GO Biological Process (2018), KEA (2015), KEGG (2019), Panther (2016), PPI Hub Proteins, Reactome (2016), Transcription Factor PPIs, and WikiPathways (2019). Only enrichments with a FDR-adjusted *p* value < 0.2 were considered statistically significant.

**Modeling of cancer cell line genetic dependencies**. Genetic dependency data were downloaded from the online DepMap[64] portal (https://depmap.org/portal/download/) (2020Q1) and the CERES scores[65] were used for all analyses. Cell lines with multiple activating *KRAS* mutations or an activating mutation in *BRAF*, *EGFR*, or *NRAS* were removed from the dataset. For each cancer, only cell lines with a *KRAS* allele found in at least three cell lines were included in the study.

The genetic dependency score is often linked to the expression of the gene. Thus, if the RNA expression of the gene could explain the dependency score (linear model, *p* value < 0.05 and $R^2 \geq 0.4$), the gene was not tested for *KRAS* allele-specific genetic dependency. Further, genes that tended to show differential dependence on the basis of their mutation status (Wilcoxon rank-sum test, *p* < 0.05) were not included in downstream analysis. Of the remaining genes, an ANOVA was used to measure if the mean dependency scores for the cell lines grouped by *KRAS* allele were different (*p* value < 0.01). For these genes, Student's *t* tests were used to compare the dependency scores of each group of cell lines against the others (Benjamini–Hochberg FDR-adjusted *p* value < 0.05). These genes were declared as differentially dependent by *KRAS* allele. The box plots in Fig. 4 and Supplementary Fig. 7 show the FDR-adjusted *p* values from the *t* tests.

**Gene set enrichment analysis of genetic dependency**. The GSEA[66] tool (version 3.0) was acquired from the online GSEA portal (https://www.gsea-msigdb.org/gsea/index.jsp). Gene sets were acquired through MSigDB (https://www.gsea-msigdb.org/gsea/msigdb/index.jsp; downloaded on 15 October 2019). The analysis used the Hallmark and C2 gene sets and permuted the genes 10,000 times for the statistical test. All other settings were set to default values. Enrichments were considered statistically significant if the adjusted *p* value < 0.2 and a normalized enrichment score < −1.2 or > 1.2.

**Modeling the effect of comutation events on genetic dependency**. For each gene found to have a genetic dependency interaction with a *KRAS* allele, an additional linear model was built to estimate the effect of mutations to genes that comutate with the *KRAS* allele. The linear model regressed on the RNA expression level of the gene and contained binary indicator variables for if the cell line had a mutation in the targeted gene, had the specific *KRAS* allele or another allele, or had a mutation in one of the genes that comutates with the specific *KRAS* allele. Only comutation genes that were mutated in at least three cell lines and WT in at least three cell lines were included as covariates. To avoid including perfectly correlated variables in the model, comutating genes that were perfectly correlated (i.e., they were mutated in exactly the same cell lines) were merged into a single predictor. After these adjustments, the models had 45, 29, and 16 coefficients for genes with dependency interactions with *KRAS* G12D, G12V, and G13D in COAD cell lines, respectively. For PAAD cell lines, the models had 15, 14, and 8 coefficients for genes with dependency interactions with G12D, G12R, and G12V, respectively. Some models had fewer covariates because either (1) the targeted gene was not mutated in enough of the cell lines to include the coefficient for this variable in the model, or (2) the targeted gene was mutated in the same cell lines as one or more of the comutating genes, resulting in the merging of these variables. Due to the imbalance between the number of covariates and data points (i.e., cell lines), the models were fit with elastic net regularization[75,105] constraining the mixing parameter $\alpha \in [0.75, 1]$, thus favoring the L1 penalty.

**Reporting summary**. Further information on research design is available in the Nature Research Reporting Summary linked to this article.

## Code availability

All code is available at https://github.com/Kevin-Haigis-Lab/kras-allele-genetic-interactions (https://doi.org/10.5281/zenodo.4541794). See the README for the

organization of the code and how to run the analyses. Python v3.7 (ref. [106]) and R v4.0 (ref. [107]) were used for most of the analyses.

## Data availability

All data that support the findings of this study are publicly available from the cited sources. The compiled data can be downloaded from FigShare (https://doi.org/10.6084/m9.figshare.14115569). The WGS, WES, and RNA expression data of COAD, LUAD, and PAAD tumor samples are available on cBioPortal (http://www.cbioportal.org). The WGS, WES, and RNA expression data of MM tumor samples are available on the Multiple Myeloma Research Foundation's Research Gateway (https://research.themmrf.org). Additional WGS and WES of PAAD tumor samples generated by the ICGC were downloaded from ICGC data portal (https://dcc.icgc.org). The panel sequencing data of tumor samples are available through the dedicated GENIE instance of cBioPortal (https://www.cbioportal.org/genie/). All users must register and agree the AACR's terms of use before accessing the data. The Cancer Gene Census data were downloaded from the COSMIC website (https://cancer.sanger.ac.uk/census). The genetic dependency data (2020Q1) and cell line WGS and RNA expression data (generated by the CCLE) were downloaded from the DepMap web portal (https://depmap.org/portal/). Normal gene expression data were downloaded from the GTEx web portal (https://www.gtexportal.org). Normal protein expression data were downloaded from the Human Protein Atlas web portal (https://www.proteinatlas.org). The remaining data are available within the Article, Supplementary Information, or Source data, or are available from the authors upon request.

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

## Acknowledgements

This work was supported by a grant from the National Institutes of Health (R01CA232372 to K.M.H.) and an award from the Cancer Research UK Grand Challenge and the Mark Foundation to the SPECIFICANCER team. The whole-exome sequencing data of MM were acquired from the Multiple Myeloma Research Foundation Personalized Medicine Initiative. The authors would like to acknowledge the American Association for Cancer Research and its financial and material support in the development of the AACR Project GENIE registry, as well as members of the consortium for their commitment to data sharing. Interpretations are the responsibility of study authors.

## Author contributions

J.H.C., G.E.M.M., P.J.P., and K.M.H. devised the research strategy. J.H.C., G.E.M.M., and D.C.G. performed the analyses. J.H.C., G.E.M.M., P.J.P., and K.M.H. wrote the manuscript. J.H.C., G.E.M.M., P.J.P., and K.M.H. helped to interpret results. All authors reviewed and approved the final manuscript.

## Competing interests

The authors declare no competing interests.
