## [Peer Review File · Nature Communications]

REVIEWER COMMENTS

Reviewer #1 (Remarks to the Author): Expert in KRas mutated lung cancers

In the manuscript entitled “The origin, distribution, and genetic interactions of KRAS alleles across cancer types”, Cook et al. implemented a series of genomic analyses to explore the mutagenic mechanism of KRAS mutations and the genetic interactions and signaling properties associated with each mutant form of K-RAS in four cancer types: colorectal adenocarcinoma, pancreatic adenocarcinoma, lung adenocarcinoma and multiple myeloma. This study addresses the genetic complexity of cancer through a comprehensive genetic interaction analyses of oncogenic KRAS alleles. Their findings support a model in which the various oncogenic KRAS mutations are not biologically redundant, but instead have distinct properties that are reflected in their genetic interactions. Most allele-specific genetic interactions were not consistent between tissues, demonstrating the complex relationship between tissue-of-origin, K-RAS function, and cooperating genetic events. This work is timely and interesting, and provides novel insights about different KRAS mutations. The study is well-designed with adequate data and appropriate statistical analyses. Below are the concerns that would strengthen the study:

(1) Authors showed that there is no difference in mutation signature patterns comparing tumors with distinct KRAS mutations (Fig 1C), the question is that whether tumors with different mutational signatures have preferred KRAS mutations? Regardless of tumor type, what is the correlation between predicted and actual KRAS mutations in tumors with each mutational signature?

(2) Figure 2 shows the predicted frequencies of KRAS alleles are different from observed frequency without a convincing explanation – please discuss/explain in some detail.

(3) Please detail how the probability of each mutational signature to have caused the KRAS mutation in a tumor sample is calculated. The statistical analysis for the association between KRAS mutations and mutational signatures needs to be provided.

(4) Is there additional evidence that supports the supposition that “The alleles never or rarely found in each cancer were predicted to occur at frequencies ranging from 1.5% (for Q61L in PAAD) to 10.5% (for Q61K in LUAD), indicating that these alleles are not rare because their causative mutations do not occur, but instead because of weak oncogenic fitness in the tissue.”

(5) When performing comutation analysis, do these mutations exist in a clonal population versus polyclonal populations?

(6) Is there any genetic/genomic evidence, such as mutual exclusivity, for these possible ‘synthetic lethal’ genes with KRAS mutations?

(7) The analysis and discussion of the relationship between tissue characteristics and specific KRAS mutations are weak and should be strengthened. For example, why does MM have more KRAS Q61H than other cancer types?

(8) The observed frequency of some KRAS alleles was greater than predicted perhaps due to positive

selection, but quite a few KRAS alleles occurred at significantly lower frequency than predicted in some cancers, possibly due to weak oncogenic fitness in the specific tissue. This issue needs to be explained and discussed.

(9) The study provided many statistically significant findings, but did not link these findings to cancer- and allele- specific differences in drug response and clinical outcome. For example, the background conveys that COAD tumors with a KRAS G13D allele are sensitive to anti-EGFR therapies; and advanced PAAD with KRAS G12D allele is associated with reduced overall survival. It would be helpful to discuss whether allele-specific mutations or differentially dependent cellular processes or other oncogenes could explain/contribute to such observations.

(10) The github code is not available with the link provided. (<https://github.com/jhrcook/comutation>). There are no supplementary tables 2-9.

Reviewer #2 (Remarks to the Author): Expert in cancer genomics

Cook et al analyzed 13492 samples from four different cancers (colorectal, pancreatic, lung and multiple myeloma), and looked for the effect of different mutant and oncogenic KRAS alleles. They showed that known mutagenic mechanisms partially explained the spectrum of KRAS alleles, but that biological selection also played a role in different tissues. They also looked for comutation networks, identifying genetic dependencies associated with specific mutant KRAS alleles.

This is an intriguing study with possible implications for KRAS driven cancers. However, the authors need to make their methods and goals far more transparent and easier for the reader to understand. Moreover, it wasn't clear if all the statistical tests (such as the comutation studies) had been adjusted for multiple testing.

Other comments:

Comutation networks to this reviewer are puzzling. One would expect that comutated genes would have been identified as genetic drivers already. The fact that many of them have not suggests that results could be spurious. Could the authors please address this issue.

Page 5: State which KRAS alleles were correlated with microsatellite instability.

Page 5, last paragraph: Reword first sentence – there are better ways of saying this.

Page 7, para, line 6: Missing words here: but “had an” actual...

Page 7, para 2, line 2: Presumably the authors are referring to KRAS alleles here – if so please insert.

Page 8, last para, line 3: Don't the authors mean known oncogenes and tumor suppressor genes? Similarly p 9, last para – TP53 is a tumor suppressor etc.

It would have been helpful for the authors to describe the actual mutations in genes where there was significant comutation (or lack of) with the KRAS alleles. For example in the case of TP53, was this affected by whether the mutation was truncating, or affected protein interaction? Similarly for the genetic dependencies from the CRISPR screens, what are the actual mutations in the genes identified (shown in Figure 4d?).

The authors report a high rate of co-mutation between KRAS Q61H alleles and NRAS in MM. Describe which allele/s of NRAS these were. Has this been reported before for MM?

Response to Reviewers

Summary

Reviewer #1

The first reviewer requested additional explanation and analysis of the associations between mutational signatures and the *KRAS* alleles. Reviewer 1 was also curious about the interpretation of our results and the characteristics of the tissue-of-origin of the tumors.

We addressed these concerns as follows:

- We conducted statistical analyses to test for differences in mutational signature compositions of tumor with different *KRAS* alleles (presented in the new Supplemental Fig. 2) and differences in the probabilities for the mutational signatures to have caused the mutation (presented in the new Supplemental Fig. 3). In addition, we assessed the extent to which the mutational signatures can be used to predict which *KRAS* allele a tumor would obtain (presented in the new panel b of Fig. 2).
- In multiple parts of the Results and Discussion, we add further interpretation of our results with regards to the tissue-specificity of *KRAS* mutations, citing several experimental studies on the topic to provide further support.

Reviewer #2

Reviewer 2 expressed concern for the potential for a high rate of false positives in the comutation networks and asked why our comutation analysis of the individual *KRAS* alleles would discover genetic interactions not recorded in previous comutation analyses. Lastly, this reviewer requested that we provide additional details about the mutations in genes found to have these comutation interactions with *KRAS*.

We addressed these concerns as follows:

- We have made an effort to clarify the goals and methods of our study throughout the manuscript. We have specifically provided a detailed explanation of the steps we used to mitigate the reporting of false positive comutation interactions and have added an additional step that filters based on an estimated false discovery rate.
- To emphasize the novelty of an allele-specific analysis, we conducted a non-allele-specific comutation analysis of *KRAS* and compared the results to the allele-specific analysis presented in the original manuscript. Briefly, the results highlight how the allele-specific analysis is crucial for identifying most of the comutation interactions of the uncommon *KRAS* alleles as they are often masked by the more common alleles.
- As requested by the reviewer, we have included additional details on the particular mutations of the genes with comutation interactions with the *KRAS* alleles.

Point-by-Point Responses

Reviewer #1

(1) Authors showed that there is no difference in mutation signature patterns comparing tumors with distinct *KRAS* mutations (Fig 1C), the question is that whether tumors with different mutational signatures have preferred *KRAS* mutations? Regardless of tumor type, what is the correlation between predicted and actual *KRAS* mutations in tumors with each mutational signature?

The question of whether tumors with different mutational signatures have preferred *KRAS* mutations is a difficult to answer directly, because each tumor has a distribution of signatures and the tumors cannot be clearly separated into different groups based on signatures. We therefore have addressed the question indirectly, by grouping tumors by their *KRAS* allele and asking whether they have different mutational signatures (Fig. 1c). In response to the reviewer's comments, we have added pairwise Wilcoxon rank-sum tests to estimate statistical significance and infer which mutational signatures stray from the expected (Supplementary Fig. 2; page 7, line 2). We should note that we have already described several clear associations between mutational signatures and their preferred *KRAS* mutations (e.g., the probability for SBS4 to have induced G12A/C/V mutations is much higher compared to G12D mutations in LUAD). We have now added a statistical test for this assertion, indicating which mutational signatures stray from the general conclusion (Supplementary Fig 3; page 7, line 20). With respect to the second question, again, we are not able to answer this question directly.

Perhaps a related question is whether if, on a per-tumor-sample basis, there is an association between the probability of each *KRAS* allele (estimated from the mutations found in the tumor sample) and the observed *KRAS* allele (the mutation actually acquired by the tumor). To address this, we analyzed the probabilities of the *KRAS* alleles in each tumor by comparing the probability of obtaining a certain *KRAS* allele between tumor samples that obtained the specific *KRAS* allele and other tumor samples (page 10, line 15). For most *KRAS* alleles, there was no difference between the probability of obtaining a specific *KRAS* mutation in tumor samples observed to have the allele compared to tumor samples with a different *KRAS* allele (Fig. 2b). The two instances where this was not the case were with G12V in COAD and G12C in LUAD, where each *KRAS* allele had a greater probability of occurring in tumor samples that actually obtained the mutation compared to other tumor samples.

(2) Figure 2 shows the predicted frequencies of *KRAS* alleles are different from observed frequency without a convincing explanation – please discuss/explain in some detail.

We have discussed this result in paragraphs four and six in “The frequency of most *KRAS* alleles cannot be solely attributed to the prevalence of detected mutagens” (page 10, line 3; page 11, line 6). We think that biological selection is driving the allele selection and, in particular, that the interactions between the distinct biological properties of each *KRAS* allele and the pre-existing signaling context of the tissue-of-origin play a key role. In other words, some *KRAS* alleles are stronger cancer drivers than others in different tissues and this, along with the active mutagenic processes, determines the frequency at which the *KRAS* alleles are observed in each cancer.

(3) Please detail how the probability of each mutational signature to have caused the *KRAS* mutation in a tumor sample is calculated. The statistical analysis for the association between *KRAS* mutations and mutational signatures needs to be provided.

The probability of each mutational signature to have caused the *KRAS* mutation in a tumor sample is explained in the Methods section “Probability of *KRAS* mutations from mutational signatures” (page 24, line 8). Some changes to this explanation have been made to clarify the method, and the revised version is copied below:

For each sample harboring a *KRAS* allele, the probability of each mutational signature to have caused the mutation was calculated by considering the weight of the base change among the 96 possibilities and the relative contribution of the signature to the mutations in the sample. Thus, the probability p of a tumor sample a to have acquired the *KRAS* mutation k from signature s of all signatures S can be calculated using Eq. 1.

$$p_{k,s,a} = \frac{c_{s,a} w_{k,s}}{\sum_i^S c_{i,a} w_{k,i}}$$

where $c_{s,a}$ is the contribution of signature s in sample a and $w_{k,s}$ is the weight of mutation k in signature s . The probability is normalized to sum to 1 by dividing by the probability of getting the observed *KRAS* mutation from any of the signatures.

The probability of a mutational signature to have caused a *KRAS* mutation was compared between two groups of tumor samples separated by their observed *KRAS* allele using a Wilcoxon rank-sum test. The p-values were adjusted for multiple hypothesis testing using the Benjamini-Hochberg method.

As stated in the response to question (1), we conducted pairwise Wilcoxon rank-sum tests to determine if there were differences in the probabilities of alleles to have been caused by each mutational signature.

(4) Is there additional evidence that supports the supposition that “The alleles never or rarely found in each cancer were predicted to occur at frequencies ranging from 1.5% (for Q61L in PAAD) to 10.5% (for Q61K in LUAD), indicating that these alleles are not rare because their causative mutations do not occur, but instead because of weak oncogenic fitness in the tissue.”

As these are rare *KRAS* mutations in the respective cancers, their oncogenic fitness has not been experimentally tested as thoroughly as the more common alleles.

We do specifically discuss the instance of *KRAS* A146T in PAAD (page 10, line 10), a very rare allele though predicted to comprise 9% of *KRAS* mutations, as we have experimental evidence of its weak fitness in this context (Poulin *et al.*, 2019). In addition, Zafra *et al.*, recently demonstrated multiple tissue-specific responses by expressing *KRAS* G12R in the colon and *KRAS* G13D in the pancreas. For instance, the expression of *KRAS* G12R in the colon was insufficient to induce the hyperplasia that is regularly observed with the more common alleles.

With specific regards to Q61 mutations, there is evidence that, provided sufficient mutagenic pressure, *KRAS* Q61L mutations can induce tumorigenesis in the lungs (Li *et al.*, 2020). In Supplementary Fig. 2, we indicate that the prediction of the frequency of Q61L is very close to the actual frequency, supporting this experimental conclusion.

(5) When performing comutation analysis, do these mutations exist in a clonal population versus polyclonal populations?

This is an interesting question that we had originally wanted to address in this study.

Nevertheless, we were unable to collect and/or calculate the variant allele fraction (VAF) data for every mutation, due to the fact that raw read-level data were not available for the panel data. As discussed in the manuscript, MM is known to be frequently multi-clonal (page 14, line 1). We presumed that not accounting for this would lead to a high-rate of false positives in the comutation analysis. Therefore, we focused on genes previously known to drive MM to avoid highlighting false positive interactions. For COAD, LUAD, and PAAD, it should be noted that, because it tends to be an early mutation, almost all of the *KRAS* mutations are clonal. Thus, all other mutation events would be in tumor cells containing the *KRAS* mutation.

(6) Is there any genetic/genomic evidence, such as mutual exclusivity, for these possible 'synthetic lethal' genes with *KRAS* mutations?

Interestingly, there was no overlap between the genes with allele-specific reduced comutation interactions and those with increased genetic dependency. We believe this to be due to two main factors. The first is that the effects of a gene not being mutated is, in most cases, distinct from complete bi-allelic loss of the gene. Thus, a CRISPR-Cas9 knockout screen does not mimic the cellular effect of mutual exclusive mutations.

The second main factor is that we do not account for gene copy number in the comutation analysis, though many collateral lethal interactions have been attributed to the loss of a single allele (often termed "copy number alterations yielding cancer liabilities owing to partial loss" or "CYCLOPS"; Nijhawan *et al.*, 2012, Muller *et al.*, 2015). Therefore, our comutation analysis would not identify many of the collateral lethality effects that would likely be identified by a genetic screen. This is a limitation of the study.

(7) The analysis and discussion of the relationship between tissue characteristics and specific *KRAS* mutations are weak and should be strengthened. For example, why does MM have more *KRAS* Q61H than other cancer types?

This is a very important open question in the field that we believe our paper contributes towards answering, though is insufficient to answer fully. Our results indicate that the distribution of frequencies of the *KRAS* alleles observed in these cancers cannot be solely attributed to passive mutational processes. Instead, we posit that the distinct biological properties of the different *KRAS* mutations contribute to their tissue-specific frequencies. This is supported by how the alleles have distinct tissue-specific genetic interactions. However, the precise biological reasons for the tissue-specificity of *KRAS* mutations has yet to be determined. Thus far, several studies from our lab and others have experimentally examined how some alleles interact with specific tissues (for example: Poulin *et al.*, 2019, Hobbs *et al.*, 2019, and Zafra *et al.*, 2020). We believe understanding these phenomena is essential to fully describing *KRAS* driven cancers and are continuing to pursue it from various perspectives.

(8) The observed frequency of some *KRAS* alleles was greater than predicted perhaps due to positive selection, but quite a few *KRAS* alleles occurred at significantly lower frequency than predicted in some cancers, possibly due to weak oncogenic fitness in the specific tissue. This issue needs to be explained and discussed.

We agree that this is an important point to address. It is discussed in paragraph four of the Results section "The frequency of most *KRAS* alleles cannot be solely attributed to the prevalence of detected mutagens" with specific reference to the high expected frequency of A146T mutations in PAAD and its experimentally demonstrated poor oncogenic fitness (page

10, line 10).

(9) The study provided many statistically significant findings, but did not link these findings to cancer- and allele- specific differences in drug response and clinical outcome. For example, the background conveys that COAD tumors with a KRAS G13D allele are sensitive to anti-EGFR therapies; and advanced PAAD with KRAS G12D allele is associated with reduced overall survival. It would be helpful to discuss whether allele-specific comutations or differentially dependent cellular processes or other oncogenes could explain/contribute to such observations.

Additional discussion of this topic has been added to the Discussion (page 20, line 5). We believe this is a very important topic and was the reason for the paper's final analysis "An integrated analysis of allele-specific comutation and genetic dependencies." The purpose of this analysis was to test whether *KRAS* allele-specific dependency interactions could alternatively be explained by mutations to genes with comutation interactions with the *KRAS* allele. This was inspired by the possibility that distinctions between *KRAS* alleles identified from ad hoc analyses of clinical data could actually be driven by mutations to other genes that happen to have comutation patterns with the *KRAS* alleles.

In addition, we have tried to statistically analyze the associations between comutation events and patient outcome, though we were limited by data availability. High-quality clinical data is relatively uncommon, thus, when grouping the patients by the *KRAS* mutation of their tumor and the mutation status of comutating genes, the sub-groups become too small to provide sufficient statistical power to identify distinctions in patient outcome.

(10) The github code is not available with the link provided. (<https://github.com/jhrcook/comutation>). There are no supplementary tables 2-9.

We apologize for the oversight on the availability of the code; the repository has been made public. For the supplementary tables, it seems that the Excel file was turned into a PDF at some point in the submission process. The entire Excel file should now be available to the reviewers.

Reviewer #2

(1) This is an intriguing study with possible implications for KRAS driven cancers. However, the authors need to make their methods and goals far more transparent and easier for the reader to understand. Moreover, it wasn't clear if all the statistical tests (such as the comutation studies) had been adjusted for multiple testing.

Per the reviewer's recommendation, we have improved the clarity of the methods and goals in the manuscript.

With specific regards to the statistical analyses of the comutation studies, we did not filter the comutation interactions based on FDR-adjusted p-values because we found these methods too strict for our purpose - removing all but the strongest interactions (e.g. the reduced comutation interactions with *BRAF* or increased comutation interactions with *APC* in COAD). Instead, we opted to use a relatively strict p-value cutoff (p-value < 0.01) and additional thresholds on other properties of the comutation interactions (such as a lower bound on the number of comutation events) to remove likely false positives. We believe this thresholding process eliminated many false positives while still providing a meaningful list of comutation interactions for further *in vitro* study. Recognizing these limitations, we chose to focus on genes known to be related to cancer or *KRAS* signaling and the results of a functional enrichment analysis to specifically highlight the comutation interactions least likely to be false positives. In this vain, we have applied an additional filter to the interactions with genes previously associated with cancer or *KRAS* signaling (those presented in Fig. 3b, Supplementary Fig. 6b and Supplementary Fig. 8b). These sets of interactions were further filtered to fall below an FDR of 0.25 that was estimated using the Benjamini-Hochberg method (page 27, line 3).

The adjustments for multiple hypothesis testing for the remaining analyses are specifically indicated in their respective Methods sections.

(2) Comutation networks to this reviewer are puzzling. One would expect that comutated genes would have been identified as genetic drivers already. The fact that many of them have not suggests that results could be spurious. Could the authors please address this issue.

The hypothesis behind this allele-specific comutation analysis is that some mutations only contribute to cancer within a specific cellular context, in this case, the unique signaling characteristics of a specific *KRAS* allele. These genes may not have been previously documented as cancer drivers as they do not behave as such on their own. Instead, they only promote cancer when accompanied by additional specific signaling perturbations.

Previous comutation studies that group all *KRAS* mutations into a single category do so under the assumption that they are identical. However, if the *KRAS* alleles are in fact distinct, then allele-specific interactions are likely to be missed, particularly for the less common alleles. Just as with any comutation analysis, interactions can be spurious, though, as explained in the previous question, we have used several methods to highlight interactions most likely to be real.

To demonstrate how the results of an allele-specific and non-allele-specific comutation analysis differ, we conducted the same statistical tests for comutation interactions currently used in the study, but now treating all *KRAS* mutations as a single group. The identified interactions were compared to those from the allele-specific analysis and are now present in Supplementary Fig. 5 and discussed in the final paragraph of "The *KRAS* alleles have distinct comutation networks" (page 15, line 8).

In COAD, LUAD, and PAAD the sets of genes found to have increased or reduced comutation interactions were substantially different between the two analyses (Supplementary Fig. 5a, c, and g). In MM, the results are quite similar (Supplementary Fig. 5e), though there were only a few interactions identified in total and there is still the factor of possible polyclonality of these tumors (as discussed in the manuscript). We have included the COAD example (Supplementary Fig 5a, b) here for illustration:

For COAD and LUAD, the interactions found from the two different analyses (allele-specific and non-allele-specific) had both considerable overlap and distinction. For example, the non-allele-specific *KRAS* comutation analysis in COAD identified 105 reduced comutation interactions, only 35 of which were also identified in the allele-specific analysis (Supplementary Fig. 5a). On the other hand, 28 novel reduced comutation interactions were only identified when the *KRAS* alleles were considered individually. The fact that 70 genes were identified only by the non-allele-specific analysis indicates that there may be some genes that comutate with several, but not all, *KRAS* alleles, and the allele-specific analysis was under-powered to identify them. Overall, the results of comparing the allele-specific and non-allele-specific analyses demonstrate that they address similar, yet distinct, biological relationships.

In PAAD, the value of an allele-specific analysis is highlighted because upwards of 90% of the tumors have a *KRAS* mutation. Therefore, a non-allele-specific analysis identified 5 reduced comutation interactions and 6 increased comutation interactions (Supplementary Fig. 5g). Conversely, the allele-specific analysis identified far more interactions, including multiple genes that simultaneously have increased comutation with some *KRAS* alleles and reduced comutation with others (Supplementary Fig. 8c).

(3) Page 5: State which *KRAS* alleles were correlated with microsatellite instability.

About 17% of COAD tumors were hypermutant. We used a one-sided Fisher's exact test to determine if any *KRAS* alleles were enriched in these tumors. Overall, hypermutant samples were more likely to be WT *KRAS* (odds ratio = 1.2, FDR-adjusted p-value < 0.05), though *KRAS* Q61K was correlated with hypermutant samples (odds ratio = 6.3, FDR-adjusted p-value < 0.001). It should be noted that Q61K is a very rare allele in COAD, found in 0.47% of the tumor samples. These results align with those previously published by Imamura *et al.* (2014, PMID: 24885062).

(4) Page 5, last paragraph: Reword first sentence – there are better ways of saying this.

The sentence, “Each mutational process is not equally likely to cause each *KRAS* allele,” has been changed to, “Each mutational process has a different propensity to induce each *KRAS* allele,” (page 7, line 20).

(5) Page 7, para, line 6: Missing words here: but “had an” actual....

The sentence, “... Q61H, which was dramatically underestimated with a predicted frequency of 15.0% but actual frequency of 35.7% of *KRAS* mutations,” was changed to, “... Q61H, which was dramatically underestimated with a predicted frequency of 15.0% but **an** actual frequency of 35.7% of *KRAS* mutations,” (page 9, line 14).

(6) Page 7, para 2, line 2: Presumably the authors are referring to *KRAS* alleles here – if so please insert.

The sentence, “correlations between the observed and predicted allele frequencies for each cancer...” was changed to “correlations between the observed and predicted ***KRAS*** allele frequencies for each cancer...” (page 9, line 18).

(7) Page 8, last para, line 3: Don’t the authors mean known oncogenes and tumor suppressor genes? Similarly p 9, last para – TP53 is a tumor suppressor etc.

The sentence, “... or are known oncogenes...” was changed to “... or are known oncogenes **or tumor suppressor genes**...” The text was also changed in similar contexts elsewhere in the manuscript (page 12, line 18; page 15, line 18; page 34, line 7; page 42, line 3; page 43, line 8).

(8) It would have been helpful for the authors to describe the actual mutations in genes where there was significant comutation (or lack of) with the *KRAS* alleles. For example in the case of TP53, was this affected by whether the mutation was truncating, or affected protein interaction? Similarly for the genetic dependencies from the CRISPR screens, what are the actual mutations in the genes identified (shown in Figure 4d?).

Brief descriptions of mutations to genes found to have comutation interactions with *KRAS* alleles have been included where relevant in the Results section “The *KRAS* alleles have distinct comutation networks” (page 11, line 14). In most cases, these mutations were those commonly found of the known cancer-associated genes (page 12, line 10). However, there were some notable trends such as the uniquely high rate of comutation between *KRAS* G12V and the *TCF7L2* R488C mutation (page 13, line 2).

We inspected the mutations to the genes with opposing comutation interactions with multiple *KRAS* alleles highlighted in Supplementary Fig. 7 (*TP53*, *RNF43*, *MAP2K4*, *RBM10*), though found no notable trends (page 14, line 21).

To further describe the interactions with known oncogenes, analyses of the comutation between the alleles of oncogenes and those of *KRAS* have been conducted. No patterns of *KRAS* alleles demonstrating differential preference for specific alleles of other oncogenes was uncovered. For instance, in the analysis of *PIK3CA* in COAD, there was a significant association of *KRAS* G12C, G12V, and G13D with *PIK3CA* Q546K and E545K for increased rates of comutation, though there were no *KRAS* alleles with reduced comutation interactions with any *PIK3CA* alleles (page 12, line 20).

A similar analysis was conducted for the comutation between *KRAS* alleles and individual protein domains of tumor suppressor genes (TSG). As above, no new interactions were identified where *KRAS* alleles demonstrated distinct patterns of comutation with particular domains of TSG.

Details of the mutation to *TP53* in the COAD cell lines and *SMAD4* in the PAAD cell lines were added to the Results section “An integrated analysis of allele-specific comutation and genetic dependencies” (page 18, lines 5-14):

“Most of the *TP53* mutations were located in the DNA binding domain, two of which were nonsense mutations. Of the other mutations, two were at splice-sites, one was in the nuclear localization signaling domain, and two more were either nonsense or frameshift mutations in the N-terminal domain. All were either predicted to be deleterious or are at hotspots previously identified by TCGA or COSMIC... All of the *SMAD4* mutations were at known COSMIC hotspots. All but two were frameshift or nonsense mutations.”

(9) The authors report a high rate of co-mutation between *KRAS* Q61H alleles and *NRAS* in MM. Describe which allele/s of *NRAS* these were. Has this been reported before for MM?

The concomitant mutations were predominantly at *NRAS* Q61 though there was no detectable pattern of comutation between specific *KRAS* and *NRAS* alleles (possibly due to the low power of the analysis caused by further subdividing the groups of tumors). This information has been added to the text in the Results section “The *KRAS* alleles have distinct comutation networks” (page 14, line 10).

The co-occurrence of *KRAS* and *NRAS* mutations in multiple myeloma have been reported previously including by Bolli *et al.* in “Heterogeneity of genomic evolution and mutational profiles in multiple myeloma” (2014, PMID: 24429703).

REVIEWERS' COMMENTS

Reviewer #1 (Remarks to the Author):

The authors have thoroughly and thoughtfully addressed the concerns of the reviewers. The study is an important contribution to the field.

Reviewer #2 (Remarks to the Author):

This is an interesting study. The authors have responded to my concerns. I don't have any others.